# Effect of Antimicrobial and Antioxidant Rich Pomegranate Peel Based Edible Coatings on Quality and Functional Properties of Chicken Nuggets

**DOI:** 10.3390/molecules27144500

**Published:** 2022-07-14

**Authors:** Sadaf Bashir, Muhammad Sajid Arshad, Waseem Khalid, Gulzar Ahmad Nayik, Sami Al Obaid, Mohammad Javed Ansari, Andres Moreno, Ioannis K. Karabagias

**Affiliations:** 1Department of Food Science, Faculty of Life Sciences, Government College University, Faisalabad 38000, Pakistan; sadafbashir.food@gmail.com (S.B.); waseemkhalid@gcuf.edu.pk (W.K.); 2Department of Food Science & Technology, Government Degree College, Shopian 192303, Srinagar, Jammu and Kashmir, India; gulzarnaik@gmail.com; 3Department of Botany and Microbiology, College of Science, King Saud University, P.O. Box 2455, Riyadh 11451, Saudi Arabia; saalobaid@ksu.edu.sa; 4Department of Botany, Hindu College Moradabad, Mahatma Jyotiba Phule Rohilkhand University, Bareilly 244001, Uttar Pradesh, India; mjavedansari@gmail.com; 5Department of Organic Chemistry, Faculty of Chemical Sciences and Technologies, University of Castilla-La Mancha, 13071 Ciudad Real, Spain; andres.moreno@uclm.es; 6Department of Food Science & Technology, School of Agricultural Sciences, University of Patras, 30100 Agrinio, Greece

**Keywords:** chicken meat, nuggets, coating, pomegranate, peels

## Abstract

The current study evaluated the effect of pomegranate peel-based edible coating on chicken nuggets in order to develop a functional and safe product, high in nutritional value. For this purpose, 2,2-diphenyl-1-picrylhydrazyl (DPPH) and total phenolic content (TPC) assays were performed to check the potential antioxidant activity of chicken nuggets; microbial control, including total aerobic count and coliforms population, was performed for quality and safety purposes; and thiobarbituric acid reactive substances (TBARS) and peroxide value (POV) were performed to determine the oxidative stability of chicken nuggets. Different treatments were applied at different storage periods (0th, 7th, 14th and 21st day). The higher value of total aerobic count (5.09 ± 0.05 log CFU/g) and coliforms (3.91 ± 0.06 log CFU/g) were obtained for the uncoated samples, while the lower population was enumerated in the combination of sodium alginate (SA) and pomegranate peel powder (PPP). However, DPPH (64.65 ± 2.15%) and TPC (135.66 ± 3.07 GAE/100 g) values were higher in the coated chicken nuggets (SA (1.5%) and PPP (1.5%)) and lowest in the control samples. The higher value of TBARS (1.62 ± 0.03 MDA/kg) and POV (0.92 ± 0.03 meq peroxide/kg) were observed in the uncoated chicken nuggets. In the Hunter color system, L*, a*, and b* peak values were determined in the coated chicken nuggets with SA (1.5%) + PPP (1.5%) at the 21st day of storage. The uncoated chicken nuggets had different sensory characteristics (appearance, color, taste, texture, and overall acceptability) compared to the coated samples. Conclusively, coating based on the combination of SA (1.5%) and PPP (1.5%) increased the quality, safety, and nutritional properties of chicken nuggets.

## 1. Introduction

Microbial contamination affects the safety and quality characteristics of foods, and may occur at any stage, during production, transport, processing, meal preparation, domestic storage, and retail trade [1,2]. Food-borne illnesses associated with pathogenic micro-organisms present a major community health concern throughout the world. Increased incidence of food borne illnesses (*Escherichia coli* O157:H7 and *Listeria monocytogenes*) has also spurred greater interest in finding innovative technologies to control infectious growth in foods. These technologies maintain quality, safety, and freshness [3]. Antimicrobials are chemical mixtures, added or naturally originate in foods that inhibit or deactivate pathogenic and spoilage microorganisms [4]. Spoilage may occur as an outcome of mishandling during the handling, commercialization, storage and distribution of the products [5].

Antimicrobial edible coating is among the novel technologies to control the bacterial development and improves protection, while delays degeneration of essence, fish and pullet products [6]. Essential oils (EOs) are fragrant and instable oily abstracts gathered from plant resources including buds, flowers, bark, roots, and verdures by revenue of appearance, fermentation, abstraction, or vapor concentration. EOs are usually applied as an additive mediator in products. These are likewise identified as a class of normal stabilizers, since their durable antimicrobial and antioxidant activity have been demonstrated in preceding inquiries [7]. In horticulture products, coating plays an important role to maintain surface moisture and firmness, and prevents the weight loss of product during storage [8]. The function of the coating is to reduce the metabolic actions in fresh food due to the reduction of respiration rate by serving as a moisture and oxygen barrier [9]. Nowadays, there are different coating components that are used, in which some polysaccharides like chitosan and alginate have gained more attention [10,11].

Pomegranate (*Punica granatum*) belongs to the *Punicaceae* family and is commonly called ‘’ponus’’ and ‘’granatus’’ that is consequent from the Latin words. The wrapping of the pomegranate characterizes practically 26–30% of the fruit. The polyphenols including flavonoids (i.e., catechin and anthocyanins), hydrolysable tannins (i.e., ellagic acid, punicalagin, gallic acid, gallic penicillin, and pedunculagin), and total antioxidant capacity are high in this portion of pomegranate. These chemical compounds are present in the pomegranate peel and juice, and justify the 92% of the antioxidant activity related with the fruit [12]. Forms of hydrolysable tannins are present in the pomegranate peel, and consist mainly of hexa-hydroxydiphenic acid and its products, as well as ellagic acid and its derivatives, penicillin, and punicalagin [13]. The pomegranate peel and seed extracts are good sources of antioxidants and antimicrobials. The efficacy of pomegranate extract (PE) is considered to be high, in scavenging the hydroxyl and superoxide anion radicals. High concentration of PE could inhibit the growth of pathogens and spoilage bacteria. The addition of PE in chicken meat products enhances its shelf life by 2–3 weeks during refrigerated storage. PE is considered effective in controlling the oxidative rancidity in chicken products [14].

Pomegranate peel holds about 40–50% of the total fruit weight [13]. It is shaped as byproduct in huge quantities by the food industry and it is an important source of bioactive compounds [15], such as hydrolysable tannins, ellagic acid, and gallic acid esters of fundamental polyol molecules [16]. The hydrolysable tannins that are found in pomegranate peel contain hexa-hydroxydiphenic critical (HHDP) and its derivatives, ellagic acid and its products, penicillin, and punicalagin [13].

Based on the aforementioned, chicken meat was used for the preparation of functional chicken nuggets using different edible coatings (SA and PPP) separately or in combination. At the 0th, 7th, 14th, and 21st days of storage intervals, DPPH, TPC, TBARS, POV, Hunter color values (L*, a*, and b*), and sensory analyses were carried out to investigate the functionality, quality, and oxidative stability of chicken nuggets.

## 2. Results

### 2.1. Microbial Analysis

#### 2.1.1. Total Aerobic Count

The total aerobic count in uncoated and coated samples of chicken nuggets during storage are given in Table 1. Microbial population was reduced with the increase of coating solution. In addition, the microbial contamination increased as the storage period increased. The total aerobic count of uncoated, SA (1.5%), PPP (1.5%), and SA (1.5%) + PPP (1.5%) formulations at refrigerated temperature (4 ± 1 °C) are shown in Table 1. The overall mean values of total aerobic count of uncoated, SA (1.5%), PPP (1.5%), and SA (1.5%) + PPP (1.5%) were 3.38 ± 0.06, 3.84 ± 0.05, 4.34 ± 0.04, and 4.70 ± 0.08 log CFU/g, respectively.

The microbial population for total aerobic count ranged between 3.05 ± 0.01 to 5.09 ± 0.05 log CFU/g. On the 21st day of storage, the microbial population for total aerobic count ranged from 3.69 ± 0.06 to 5.09 ± 0.05 log CFU/g. During storage at refrigerated temperature, a slight changes in total aerobic count was determined in the coated samples. An extreme value of total aerobic count (5.09 ± 0.05 log CFU/g) was recorded in uncoated sample at 21st day of storage, whereas the least value of total aerobic count (3.05 ± 0.01 log CFU/g) was obtained with the coating of SA (1.5%) and PPP (1.5%) on 0th day. The present findings showed that the total aerobic count was affected significantly (*p* < 0.05) by the coating formulations and storage time.

#### 2.1.2. Coliforms

The coliforms value of chicken nuggets at different storage periods are given in Table 1. The findings showed that the coliforms count varied significantly with respect to the coating formulations and storage time. The coliforms values of uncoated, SA (1.5%), PPP (1.5%), and SA (1.5%) + PPP (1.5%) formulations at refrigerated temperature (4 ± 1 °C) are shown in Table 1. The mean coliforms values of uncoated, SA (1.5%), PPP (1.5%), and SA (1.5%) + PPP (1.5%) were 1.62 ± 0.02, 1.91 ± 0.09, 2.26 ± 0.07 and 3.26 ± 0.05 log CFU/g, respectively.

The microbial population of coliforms ranged from 1.46 ± 0.04 to 2.42 ± 0.06 log CFU/g in the beginning of the experiment. On the 21st day of storage, the microbial population for coliforms ranged from 1.74 ± 0.02 to 3.91 ± 0.06 log CFU/g. Minor reinstate in coliforms was observed in the coated samples (in the mean values) during refrigerated storage. A high value of coliforms (3.91 ± 0.06 log CFU/g) was recorded on the 21st day of storage, although a minor coliforms population of 1.46 ± 0.04 log CFU/g was obtained for the coating of SA (1.5%) and PPP (1.5%) on 0th day. Therefore, present results showed that the coliforms population increased significantly (*p* < 0.05) in the uncoated as compared to coated chicken nuggets.

### 2.2. Stability of Chicken Nuggets

#### 2.2.1. Thiobarbituric Acid Reactive Substances (TBARS)

Thiobarbituric Acid Reactive Substances control is another methodology that is normally used as a guide of rancidity of fatty foods during storage [17]. The TBARS values of in the uncoated and coated chicken nuggets (SA (1.5%), PPP (1.5%), and SA (1.5%) + PPP (1.5%) formulations) are shown in Table 2. The results showed that the TBARS values were affected by the coating formulations and storage time. The mean values of uncoated, SA (1.5%), PPP (1.5%), and SA (1.5%) + PPP (1.5%) formulations were 0.71 ± 0.05, 0.80 ± 0.05, 1.03 ± 0.04, and 1.32 ± 0.03 MDA/kg, respectively. The TBARS values on 0th day ranged from 0.51 ± 0.04 to 0.91 ± 0.02 MDA/kg, individually, whereas at the end of the experiment (21st day) the TBARS values ranged from 0.86 ± 0.02 to 1.62 ± 0.03 MDA/kg. Higher TBARS values (1.62 ± 0.03 MDA/kg) were obtained for the uncoated chicken nuggets during refrigerated storage, while significantly lower TBARS values (0.51 ± 0.04 MDA/kg) were obtained for the coated chicken nuggets with SA (1.5%) and PPP (1.5%) on 0th day. As an overall result, TBARS values were affected by the coating formulations and storage time. Despite this, there was no significant (*p* > 0.05) difference between SA and PPP during refrigerated storage. However, the amount of PPP (1.5%) and SA (1.5%) + PPP (1.5%) showed a good efficiency in preventing lipid oxidation in the coated chicken nuggets.

#### 2.2.2. Peroxide Value (POV)

POV basically measures the number of peroxides that develop in meat products due to auto-oxidation. During processing, the oxidation process starts due to the auto-oxidation of unsaturated fats. The results of auto-oxidation produce off essences and off-odors that directly affect the quality of products.

The peroxide values of uncoated, SA (1.5%), PPP (1.5%), and SA (1.5%) + PPP (1.5%) formulations at refrigerated temperature (4 ± 1 °C) are shown in Table 2. The overall mean values of peroxide values of uncoated, SA (1.5%), PPP (1.5%), and SA (1.5%) + PPP (1.5%) at refrigerated temperature for 0th, 7th, 14th, and 21st day were 0.52 ± 0.04, 0.60 ± 0.02, 0.67 ± 0.05, and 0.78 ± 0.03 meq peroxide/kg, respectively. The peroxide values on 0th day ranged from 0.42 ± 0.03 to 0.61 ± 0.06 meq peroxide/kg, while at the end of the experiment (21st day) the values ranged from 0.63 ± 0.01 to 0.92 ± 0.03 meq peroxide/kg. Higher peroxide values (0.92 ± 0.03 meq peroxide/kg) were recorded for the uncoated samples during refrigerated storage, while minimum values for POV (0.42 ± 0.03 meq peroxide/kg) were obtained for the SA (1.5%) and PPP (1.5%) on 0th day. Present results showed that peroxide values increased significantly (*p* < 0.05) in the uncoated chicken nuggets samples with respect to storage time.

### 2.3. Antioxidants Potential

#### 2.3.1. Total Phenolic Contents (TPC)

The TPC of uncoated, SA (1.5%), PPP (1.5%), and SA (1.5%) + PPP (1.5%) at refrigerated temperature are given in Table 3. The overall mean values of TPC of uncoated, SA (1.5%), PPP (1.5%), and SA (1.5%) + PPP (1.5%) were 101.49 ± 2.15, 106.87 ± 2.75, 113.30 ± 3.13 and 125.68 ± 3.17 mg GAE/100 g, respectively. The TPC of chicken nuggets among different coating treatments ranged between 112.36 ± 2.03 to 135.66 ± 3.07 mg GAE/100 g on 0th day, whereas at the 21st day of storage, the TPC ranged from 91.38 ± 2.15 to 117.68 ± 2.45 mg GAE/100 g. Higher TPC values (135.66 ± 3.07 mg GAE/100 g) were obtained for the SA (1.5%) and PPP (1.5%) samples on 0th day, while minimum values for TPC (91.38 ± 2.15 mg GAE/100 g) were recorded in the control samples during the 21st day of refrigerated storage. Present results showed that TPC increased significantly (*p* < 0.05) on the coated (SA (1.5%) + PPP (1.5%)) chicken nuggets compared to the uncoated ones, during storage under refrigeration. This finding is due to the high amount of phenolic compounds in the coated samples, given that the coating materials used contain rich phenolic compounds. For instance, sodium alginate contains glucuronic acid and monotonic acid that develop a layer. This layer aids in the reduction of microorganisms population and prevents hydration of the product.

#### 2.3.2. Diphenyl-1-Picrylhydrazyl (DPPH)

The antioxidant activity of chicken nuggets based on the DPPH assay in regard to the treatments (uncoated, SA (1.5%), PPP (1.5%), and SA (1.5%) + PPP (1.5%) formulations) and storage time is shown in Table 3. The DPPH inhibition of uncoated, SA (1.5%), PPP (1.5%), and SA (1.5%) + PPP (1.5%) were 49.76 ± 1.40, 54.22 ± 1.50, 56.83 ± 1.60 and 60.94 ± 1.70%, respectively. The DPPH inhibition on 0th day ranged from 55.34 ± 1.00 to 64.65 ± 2.15%, respectively, whereas at the 21st day of storage, the DPPH inhibition in the uncoated chicken nuggets samples decreased (43.48 ± 1.60%). Higher DPPH (64.65 ± 2.15%) inhibition was obtained for the SA (1.5%) and PPP (1.5%) samples on 0th day, and the minimum DPPH inhibition (43.48 ± 1.06%) was recorded for the uncoated chicken nuggets samples on 21st day. Present results showed that DPPH inhibition was significantly (*p* < 0.05) affected by the coating formulations at different storage time.

### 2.4. Hunter Color

The Hunter color scale is a quality parameter of food products on the basis of their developed color. During coating, the color of products depends upon the ingredient that are used for coating. The color values of uncoated, SA (1.5%), PPP (1.5%), and SA (1.5%) + PPP (1.5%) formulations at refrigerated temperature are shown in Table 4 and Table 5. The L* values of uncoated, SA (1.5%), PPP (1.5%), and SA (1.5%) + PPP (1.5%) were 59.03 ± 1.84, 59.24 ± 1.85, 60.55 ± 2.16 and 61.93 ± 2.02, respectively. The a* values of uncoated, SA (1.5%), PPP (1.5%), and SA (1.5%) + PPP (1.5%) during refrigerated storage for 0, 7, 14, and 21st day were 11.75 ± 0.47, 12.69 ± 0.52, 13.98 ± 0.58, and 14.68 ± 0.62, respectively. Similarly, the b* values of uncoated, SA (1.5%), PPP (1.5%), and SA (1.5%) + PPP (1.5%) during refrigerated storage for 0th, 7th, 14th, and 21st day were 9.56 ± 0.17, 10.47 ± 0.22, 10.69 ± 0.02, and 10.87 ± 0.24, respectively.

Higher L* values (62.36 ± 2.15) were obtained for the chicken nuggets coated with SA (1.5%) + PPP (1.5%) at 21st day of storage, whereas the lower L* values (58.12 ± 1.06) were obtained for the uncoated chicken nuggets on 0th day. Present findings showed that L* values were significantly (*p* < 0.05) affected by non-coating and storage time. The L* values of chicken nuggets increased with respect to the coating formulations and storage time. The coated samples had higher L* values as compared to the uncoated samples. Increment in the L* values may be due to the high antioxidants content in the used coating materials. Higher a* values (14.87 ± 0.62) were obtained for the chicken nuggets coated with SA (1.5%) + PPP (1.5%) at 21st day of storage, while the minimum a* values (11.36 ± 0.45) were obtained for the uncoated chicken nuggets on 0th day. However, the present results showed that a* values were affected significantly (*p* < 0.05) by non-coating and storage time. The a* values of chicken nuggets were significantly different. The coated samples had higher a* values as compared to the uncoated samples. These differences are probably due to the coating material used (SA and PPP). Higher b* values (11.25 ± 0.34) were obtained for the chicken nuggets coated with SA (1.5%) + PPP (1.5%) on the 21st day of storage, while the minimum b* values (9.15 ± 0.34) were obtained for the uncoated chicken nuggets on 0th day. During storage, the b* values of coated chicken nuggets were found to be enhanced as compared to the uncoated samples. The increments in the present results may be due to the oxidation process that starts during storage. Moreover, results showed that b* values of coated chicken nuggets increased significantly (*p* < 0.05) with respect to the non-coating samples, due to the different composition of the used coating materials.

### 2.5. pH

The pH values of uncoated, SA (1.5%), PPP (1.5%), and SA (1.5%) + PPP (1.5%) formulations during refrigerated storage are shown in Table 5. Results showed that the pH values of chicken nuggets samples were significantly affected by the coating formulations at different storage time. The pH values of uncoated, SA (1.5%), PPP (1.5%), and SA (1.5%) + PPP (1.5%) were 6.19 ± 0.08 to 6.24 ± 0.23, respectively. The pH values of chicken nuggets on 0th day ranged from 5.89 ± 0.01 to 6.1 ± 0.30, whereas the pH values on 21st day ranged from 6.42 ± 0.25 to 6.45 ± 0.34, respectively. Higher pH (6.45 ± 0.34) was obtained for the uncoated chicken nuggets samples on 21st day and lower pH (5.89 ± 0.01) for the chicken nuggets samples coated with SA (1.5%) and PPP (1.5%) on 0th day. Present results showed that pH increased significantly (*p* < 0.05) with respect to the coating formulations and storage time.

### 2.6. Sensory Evaluation

#### 2.6.1. Appearance

The appearance score values of chicken nuggets for the different treatments are shown in Table 6. The score values of appearance of uncoated, SA (1.5%), PPP (1.5%), and SA (1.5%) + PPP (1.5%) were 6.27 ± 0.11, 6.24 ± 0.09, 6.48 ± 0.17 and 6.69 ± 0.28, respectively. The appearance score values of chicken nuggets among the different coating formulations ranged between 7.36 ± 0.14 to 7.89 ± 0.33 on 0th day, while at the end of experiment (21st day) the appearance score values ranged from 5.55 ± 0.17 to 5.98 ± 0.38. Higher appearance score values (7.89 ± 0.33) were obtained for the uncoated chicken nuggets samples on 0th day during refrigerated storage, while the minimum appearance score values (5.29 ± 0.10) were obtained for PPP (1.5%) chicken nuggets samples on 21st day. Present results showed that appearance decreased significantly (*p* < 0.05) with respect to the coated chicken nuggets and storage time.

#### 2.6.2. Texture

The mean ± SD texture score values of chicken nuggets with respect to the different treatments during refrigerated storage are shown in Table 6. The overall score values of texture of uncoated, SA (1.5%), PPP (1.5%), and SA (1.5%) + PPP (1.5%) were 6.51 ± 0.19, 6.59 ± 0.25, 6.57 ± 0.21, and 7.01 ± 0.02, respectively. The texture score values of chicken nuggets among the different coating formulations were found to be in the range of 7.89 ± 0.35 to 8.12 ± 0.30 on 0th day, while at the end of the experiment the values ranged from 5.49 ± 0.15 to 6.1 ± 0.02. Higher texture (8.12 ± 0.30) score values were obtained for the uncoated chicken nuggets samples on 0th day, while the minimum score values for texture (5.49 ± 0.15) were obtained for the PPP (1.5%) chicken nuggets samples on 21st day of refrigerated storage. Present results showed that appearance decreased significantly (*p* < 0.05) with respect to the uncoated chicken nuggets at different storage time.

#### 2.6.3. Taste

The taste score values of chicken nuggets with respect to the different treatments during refrigerated storage are shown in Table 7. The taste score values of uncoated, SA (1.5%), PPP (1.5%), and SA (1.5%) + PPP (1.5%) were 6.48 ± 0.12, 6.61 ± 0.18, 6.50 ± 0.13 and 6.70 ± 0.22, respectively. The taste score values of chicken nuggets ranged between 7.25 ± 0.25 to 7.75 ± 0.27 on 0th day, while at the end of the experiment the taste score values ranged from 5.68 ± 0.10 to 5.76 ± 0.12. Higher taste score values (7.75 ± 0.27) values were obtained for the uncoated chicken nuggets samples on 0th day during refrigerated storage, while the minimum score values for taste (5.68 ± 0.10) were obtained for the uncoated chicken nuggets on 21st day. The current results showed that taste decreased significantly (*p* < 0.05) in regard to the coating formulations and storage time.

#### 2.6.4. Odor

The odor score values of the chicken nuggets during refrigerated storage are shown in Table 7. The odor score values of uncoated, SA (1.5%), PPP (1.5%), and SA (1.5%) + PPP (1.5%) chicken nuggets samples were 6.38 ± 0.24, 6.41 ± 0.27, 6.84 ± 0.29, and 6.52 ± 0.21, respectively. The odor score values of chicken nuggets among the different coating formulations were found to be in the range of 7.2 ± 0.20 to 7.48 ± 0.26 on 0th day, while at the end of the experiment the odor score values ranged from 5.72 ± 0.04 to 5.88 ± 0.06. Higher odor score values (7.54 ± 0.30) were obtained for the PPP (1.5%) coated samples on 0th day during refrigerated storage, while the minimum odor score values (5.72 ± 0.04) were obtained for the uncoated chicken nuggets samples on 21st day.

#### 2.6.5. Overall Acceptability

The overall acceptability of uncoated chicken nuggets samples, SA (1.5%), PPP (1.5%), and SA (1.5%) + PPP (1.5%) during refrigerated storage are given in Table 8. The overall acceptability score values of uncoated, SA (1.5%), PPP (1.5%), and SA (1.5%) + PPP (1.5%) chicken nuggets samples were 6.15 ± 0.22, 6.35 ± 0.23, 6.43 ± 0.23, and 6.87 ± 0.24, respectively. The overall acceptability score values of chicken nuggets among the different coated treatments were found to be in the range of 6.99 ± 0.24 to 7.69 ± 0.27 on 0th day, while at the end of experiment the values ranged from 5.69 ± 0.20 to 6.2 ± 0.27. Higher overall acceptability (7.69 ± 0.27) score values were obtained for the uncoated samples during refrigerated storage on 0th day, while the minimum values for the overall acceptability (5.69 ± 0.20) were obtained for the coated chicken nuggets samples with SA (1.5%) + PPP (1.5%) on 21st day. The current results showed that the overall acceptability decreased significantly (*p* < 0.05) in regard to the coating formulations and storage time.

## 3. Discussion

Different plant sources comprise a vital source of chemical compounds, including antimicrobials and antioxidants and total phenolic compounds. The peel of fruits contains these chemicals in abundant quantities. However, the pomegranate peel is a rich source of bioactive chemicals that play valuable role in the preservation of food products. In the current study sodium alginate and pomegranate peel powder were used separately and in combination as coating materials. Results showed the potential antimicrobial and antioxidant properties of the coating formulations used for the chicken nuggets. In another study, a natural antimicrobial and nontoxic substance called chitosan was used as a coating material, comprising however, a double practical component on the stability and shelf-life extension of foods. Chitosan plays also a significant role in the inhibition of fungus progression, by decreasing the fungus cell wall structure of protein inhibitors [18].

Moosavi-Nasab et al. [19] carried out a study on fish fillet quality and safety in which two types of the coating (chitosan coating and chitosan coating laterally with black pepper oil) were used during the storage of the product under refrigeration. The antimicrobial activity of chitosan was also shown in another study when applied to fresh fruit as an edible coating without additional antimicrobial agents [20]. Previous studies have reported that chitosan antifungal properties are due to a motivation of defense enzymes [21].

Green tea extracts (rich in antioxidants and polyphenols) were used in food to prevent the oxidation process in triglycerides [22]. Lipid oxidation can be invented and improved by dissimilar appliances, comprising the invention of singlet oxygen, enzymatic and non-enzymatic formation of free radicals and active oxygen [23]. The alginate based film layers on the shallow of the product may delay oxygen diffusion and, thus, might have undersized fat rancidities [22].

GSE (grape seed extract) is labeled as a strong antiradical substance possessing scavenging actions beside free fundamental and Schiff-base metal complexes, as well as synergistic achievement with other antioxidants [22]. Yingyuad et al. [24] reported that chitosan coating could efficiently preserve refrigerated grilled pork in contradiction of lipid oxidation (which was totally expected) in agreement with the results of the present study. Yerramilli, [25] compacted β-carotene by soybean protein isolate (SPI), sodium caseinate (SC), and whey protein isolate (WPI) by the homogenization-evaporation technique, and reported that that cellular antioxidant action of SC, WPI, and SPI nanoparticles was higher (60%) compared to β-carotene alone (45%) in CaCO_2_ cells.

A previous study indicated that edible coating of cashews reduces lipid oxidation by protecting them from oxygen exposure during storage [26]. During the storage, an additional protective effect against lipid oxidation of the studied samples treated with GSE is probably related to the development of phenolic aldehydes, due to squalor of some phenolic compounds [27]. The results of our study (TBARS analysis) may be connected to those reported in a previous work for chicken breast meat [28].

The current results of the POV showed that significant differences (*p* < 0.05) were occurred in both coated and uncoated samples of chicken nuggets. The coated materials PPP and SA + PPP are vital sources of antioxidants that resulted in the lower POV in the coated chicken nuggets samples as compared to the uncoated ones. These results are in agreement with those of a previous study [29]. During storage, the value of peroxides increased, but lower POV values were obtained for the coated chicken nuggets samples. This is probably related to the antioxidant mechanisms within the coated medium structure. Antioxidants lead to the construction of quality in the product by creating cross-linking pathways; antioxidants decrease the distribution of oxygen in the surface of the product [30].

During refrigerated storage, similar findings were reported in previous studies for saline raw minced chicken meat and raw chicken meat [29,30]. In the current study, the antioxidants from the coated sources decreased the POV of chicken nuggets. These results are in line with those reported by Jeon et al. [31], who used chitosan coatings in Atlantic cod samples. In another study, whey protein and soy protein coatings were applied on sausages and beef, respectively, during refrigerated storage to evaluate the products’ safety and quality. The results of this study [32], are similar to the current findings. Topuz et al. [33] reported that pomegranate juice has a high amount of phenolic compounds. In this context, the phenolic-rich coating including *Zataria multiflora* essential oil and chitosan was used to delay/decline oxidation in pomegranate juice [34].

Previous findings showed that fruits and vegetables are the sources of different phenolic compounds; these compounds are basically secondary metabolites that act as antioxidants during the process of oxidative stress [33]. Phenolic compounds also play an important role in the auto-oxidation and chelation of metal ions that can modulate the activity of enzymes. In a previous study, in which alginate was mixed with carvacrol and methyl cinnamate, it was reported the preparation of relevant coating materials for fruits. Some other similar studies, in which some coating materials were used for various fruits like chitosan coating for strawberries; chitosan and alginate coating for blueberries and alginate coating for sweet cherry [35,36,37] are in line with the present work.

The results of the present study are in agreement with those reported by Krishnamoorthy et al. [38], who documented that coating can protect the total phenolic compounds in apple slices. Another study reported that standalone films have antioxidant activity. The test film was rich in natural volatile antioxidants that prevented oxidation [39]. In addition, the present results are in agreement with Zhang et al. [40], who added vinegar into the fatty portion of the studied product. The obtained peroxide values declined significantly in the treated samples. In fruits, there are different phytochemical profiles as reported by Kim et al. [41], who showed that fruit scavenging activity was influenced by the flavonoid content. Mohammadian et al. [42] conducted a study in which whey protein nanofibrils (WPNFs) were used. The results showed that WPNFs exhibited DPPH radical-scavenging activity. However, browning is well related to radical scavenging activity. Another study showed that WPNFs have a high radical-scavenging activity that inhibits the oxidation process in food products [43].

The alginate-based coating chicken nuggets samples recorded higher color values as compared to the uncoated samples. This finding is probably related to the properties of polysaccharides, which are surface browning materials, the oxidative rancidity and dehydration. These observations are related to the results reported by Chidanandaiah and Sanyal [44]. In the current study, the color values of the chicken nuggets decreased with the increase in storage time intervals. These variations in color are due to the beginning of oxidation and microbial population growth. Moreover, Garcia et al. [45] reported that there were no variations in color when potatoes were stored under edible coatings. In another study [6], fish fillets were stored under cinnamon coatings. The reported results indicated higher b* values (yellowness) in fish fillets, which is in agreement with the results of the present study. A previous study was carried out on poultry meat in which a mixture of thyme oil and chitosan were used as the coating agents. The results showed that there was no difference in b* values in all treatments [46]. Keokamnerd et al. [47] reported that a* values of ground chicken meat decreased during refrigerated storage. Correspondingly, the pepper color significantly decreased when the used coatings were made with a combination of chitosan and lemongrass oil [48]. During processing, the meat pigment called myoglobin is oxidized and creates discoloration in meat [49]. Feng et al. [50] reported that during the storage of fish, some continuous changes occur including resilience, chewiness, hardness and springiness. These forms are due to protein, microbial activities, and endogenous enzymes activity in muscles and fat [51].

In the current study, the pH values were slightly different for the uncoated and coated chicken nuggets during refrigerated storage. Moreover, the pH values of chicken nuggets were decreased. This finding may be probably due to the effect of coating material (sodium alginate). These findings are in agreement with the results reported in previous studies [52,53]. Duan et al. [54], reported in an earlier study that the pH values of chicken meat were basically constant at different storage times when the products were coated with chitosan. Similar findings were reported in previous studies dealing with chicken bone [55,56], sausages, or chicken patties treated with chitosan and essential oil [57].

Dashti et al. [58] reported that with the addition of a natural source of antioxidants, the appearance of nuggets differentiated. This finding is in agreement with the results of the present study. In another study, dealing with beef patties, sodium alginate used as a coating material resulted in a decrease in the appearance score values [59]. During the refrigerated storage of the coated and uncoated chicken nuggets, there were monitored differences in the texture of samples in agreement with the results reported in previous studies [52,60]. Horita et al. [61] carried out a study on meat products in which the capacity of holding water and fat through the better development stage during different storage intervals resulted in the differentiation of the meat texture. In the current results, the decrease in the flavor score values of chicken nuggets during refrigerated storage may be probably related to the decrease in the unpredictable flavor segments, given the fat oxidation that takes place. Drake and Drake [62] reported that refrigerated chicken patties had decreased taste score values. Similarly, Sarower et al. [63] reported that the taste and flavor of nuggets diminished essentially with the development of storage time intervals. The authors clarified that this phenomenon might be due to the lipid peroxidation in the nuggets, which results in the decrease in the taste and flavor scores of the samples. In the results of odor, the obtained score values are related to the coating materials and storage time. Wood et al. [64] reported that the shrinking in flavor scores might be due to the peroxidation of polyunsaturated fatty acids that leads to the development of rancid flavor and odor. Comparable results were also reported in previous studies dealing with food products [52,60,65] stored under refrigeration. The overall acceptability score values of chicken nuggets changed significantly with respect to storage time and coatings. The general dullness scores of coating with PPP (1.5%) and SA + PPP preparation were rather lower than coating with SA (1.5%) formulation. Giatrakou et al. [46] stated that chitosan coating applied on chicken resulted in differences in the acceptability score values of the product in relation to coated or uncoated samples. In another study, in agreement with the present results, the overall acceptability values of mutton patties were differentiated during storage time [66,67,68].

## 4. Materials and Methods

### 4.1. Procurement of Raw Material, Chemicals and Reagents

Pomegranate peel powder (PPP) and sodium alginate (SA) were used for the preparation of the edible coating of chicken nuggets. Physicochemical, microbiological, and antioxidant parameter analyses of the product were determined at storage intervals of 0th, 7th, 14th, and 21st day. Chicken meat was procured from local market of Faisalabad, Pakistan. Pomegranate peels were also collected from local markets and were dried. After the drying period, the dried peels were converted into powder and stored in air-tight containers for further analysis. All chemicals and reagents were purchased from Sigma Aldrich (Tokyo, Japan).

### 4.2. Formulation of Chicken Meat Nuggets

Chicken meat nuggets were prepared following the method of Perlo [69]. Firstly, chicken was washed, cleaned, and weighted according to the method. Tap water was used to wash the raw meat chicken, and an electric mincer (Model; MINI-12-F, Monterrey, Mexico) was used to mince the raw chicken for the formation of high textured nuggets. A meat mixer was used for the grinding of the raw meat and onions for 5 min, and then all the other components were added, according to the method [69]. According to recipe, all raw materials were cleaned and weighed properly for the formation of chicken nuggets. The following ingredients were used in the recipe: chicken meat (500 g), plain flour (120 g), egg (1), black pepper (12 g), breadcrumbs (70 g), onion (1), oil (as required for frying), garlic paste (1tsp), and salt (20 g). After nuggets formation, these were coated with 3 different type of coating materials (1.5% SA, 1.5% PPP, and 1.5% SA + 1.5% PPP). Afterwards, chicken nuggets were immersed separately in breadcrumbs and plain flour then fried in canola oil at 180 °C until a golden-brown color of the chicken nuggets was achieved. Then, the coated nuggets were stored under refrigeration. The nuggets were then subjected to physicochemical, sensory, and microbiological analyses at the 0th, 7th, 14th, and 21st day.

### 4.3. Application and Preparation of Coating Solutions

The experimental product was coated with 1.5% solution of SA, PPP and SA+ PPP Figure 1. Firstly, a magnetic stirrer was used to mix the SA by continuous stirring to inhibit the hydroxylation at 80–90 °C. After that plasticizer agent (glycerine) was added, and hydrolysed solution was cooled at 70 °C. After cooling, 1.5% solution of PPP was added in solution. The coated solution was dipped in the 2% solution of salt (CaCl_2_) for 1 min. After the coating of nuggets, these were kept in a hot air oven at 40 °C for 30 min for the efficient casting of coating over the nuggets. The nuggets were coated with SA (1.5%), PPP (1.5%) and SA (1.5%) + PPP (1.5%) and packed properly, along with the uncoated nuggets which served as the control samples. These were labeled and kept at 4 ± 1 °C. This experimental product was evaluated at different storage time intervals (0th, 7th, 14^th^, and 21st day), on the basis of physicochemical, sensory, and microbiological analyses.

### 4.4. Microbiological Analysis

#### Total Aerobic Count and Coliforms

Microbial load (total aerobic bacteria and coliforms) of different treated chicken nuggets were determined according to the method described by Helrich [70]. Chicken nuggets samples were put into individually into the augmentation broth and incubated at 37 °C for 24 h. After that, the microbial counts were characterized and measured by using the colony forming units.

### 4.5. Physicochemical Analysis

#### 4.5.1. Thiobarbituric Acid Reactive Substances (TBARS)

Chicken nugget samples (5 g in 15 mL of distilled water) were standardized (1130× *g*) for 60 s. Then, 1 mL of the standardized solution was transferred to a test tube and the estimation of the lipid oxidation was determined by using the method of Ahn et al. [71]. Briefly, 50 µL of 7.2% butylated hydroxyanisole (BHA), L of 20 mM solution of thiobarbituric acid (TBA) and 15% trichloroacetic acid (TCA) solution were added in a test tube. Water bath was used for the heating of the test tubes at 90 °C for 25–30 min. Moreover, these tubes were cooled and centrifugation (2090× *g*) for 13–15 min and kept in room temperature. Spectrophotometer (Irmeco, u2020 Lütjensee, Germany) was used to measure the absorbance of the nuggets at 532 nm. TBARS values were estimated by measuring by using the following equation:Malonic dialdehyde (mg/kg meat) = (sample absorbance − blank) × Total sample volume/0.000156 × 1000. 

#### 4.5.2. Peroxide Value (POV)

The chicken nuggets samples were used to measure the peroxide value. The peroxide value was determined according to the method suggested by International Dairy Federation (IDF), as described by Koniecko [72]. Spectrophotometer (Irmeco, u2020 Lütjensee, Germany) was used to measure the absorbance of the nuggets at 500 nm. The POV results were expressed as milli- equivalents (meq)/kg of chicken nuggets, considering the following equations:Peroxide Value = (A_s_ − A_b_) × m/55.84 × m_0_ × 2 

A_s_ = Absorbance of the sample, A_b_ = Absorbance of the blank, M (Standard) = 41.52, m_0_ = mass in gram, 55.84 = atomic weight of iron.

### 4.6. Antioxidant Potential of Chicken Nuggets

#### 4.6.1. Total Phenolic Content (TPC)

Total phenolic content of chicken nuggets (uncoated and coated) was determined according to the method of Tezcan and Sever, [73]. The samples (125 µL nuggets, 500 µL ethanol (95%), distilled water (2.5 mL), and the Folin-Ciocalteu reagent (250 µL)) were prepared and added in a test tube. After 5 min, 500 µL of Na_2_CO_3_ (5%) was added and the mixture was vortexed. Then, the test tube was kept in a dark room for 1 h. Spectrophotometer (Irmeco, u2020 Germany) was used to measure the absorbance at 725 nm. Total phenolic content was expressed as gallic acid equivalents using a standard gallic acid curve (mg of GAE/g).

#### 4.6.2. Diphenyl-1-Picrylhydrazyl (DPPH)

The DPPH inhibition (antioxidant activity) of the chicken nuggets was estimated according to the method of Brand-Williams et al. [74]. The chicken nugget solution (125 µL) was mixed with 0.0012 mM DPPH solution followed by the addition of 95% MeOH up to a final volume of 4 mL and left for 1 h at room temperature in a dark place. Then the absorbance was measured at 517 nm. Antioxidant activity was determined using the following equation:Antioxidant activity (%) = 100 × (A _blank_ − A _sample/_A _blank_)

### 4.7. pH Measurement

The pH of chicken nuggets samples was measured with a pH meter (Model 520A, Orion Research inc., Boston, MA, USA). Distilled water (50 mL) was mixed with chicken nuggets (10 g) and the pH-meter was immersed to the obtained solution to measure the pH value.

### 4.8. Hunter Color

The color of chicken nuggets samples was measured with a Hunter colorimeter (Chromameter, CHROMA-400, Pulsed xenon lamp, 400 nm to 700 nm, Tokyo, Japan) at storage intervals (0th, 7th, 14th, and 21st day). The chicken nuggets were kept in a plate under the photocell. The color of chicken nugget samples surface was determined by the Hunter colorimeter with measurements standardized with regard to a white calibration plate (L* = 89.2, a* = 0.921, and b* = 0.783). CIEb * (yellowness), CIEL * (lightness), and CIEa * (redness) are the average values of 8 random reads of different viewing apertures of the samples.

### 4.9. Sensory Evaluation

Sensory evaluation of the different treated chicken nuggets samples was conducted on the basis of color, flavor, texture, taste, and overall acceptability following the procedure of Meilgaard et al. [75]. The trained panelists tested the chicken nuggets and the results were recorded at room temperature. Water was given to all experienced panelists to rinse their mouths between the samples. The panelists evaluated the chicken nuggets and their results were concluded on a score sheet (9-point hedonic scale).

### 4.10. Statistical Analysis

The obtained data were subjected to statistical analysis by applying a complete randomized design (CRD). Level of significance (*p* < 0.05) was determined by applying analysis of variance (ANOVA), following the principles outlined by Steel and Torrie [76]. The mean values were compared by using the least significant difference (LSD).

## 5. Conclusions

It is concluded that the pomegranate peel-based edible coating influences the safety and the quality of chicken nuggets during refrigerated storage. The higher values of TBARS and POV found in the uncoated chicken nuggets shows that the coating formulations used are more stable. The pH values of chicken nuggets increased with respect to storage time. However, the uncoated samples had higher pH values compared to the coated ones. The results of DPPH and TPC were significantly higher in the coated (SA (1.5%) and PPP (1.5%) chicken nuggets and lowest in the uncoated samples. The minimum values of total aerobic count and coliform were found in the combination of SA (1.5%) and PPP (1.5%). In the coated chicken nuggets, higher L*, a*, and b* values were found at the end of storage, whereas acceptable sensory parameter values (appearance, color, taste, texture, and overall acceptability) were also observed in the coated chicken nuggets. Therefore, edible coating consisted of SA (1.5%) and PPP (1.5%) may provide better quality and safety characteristics for chicken nuggets during refrigerated storage.

## Figures and Tables

**Figure 1 molecules-27-04500-f001:**
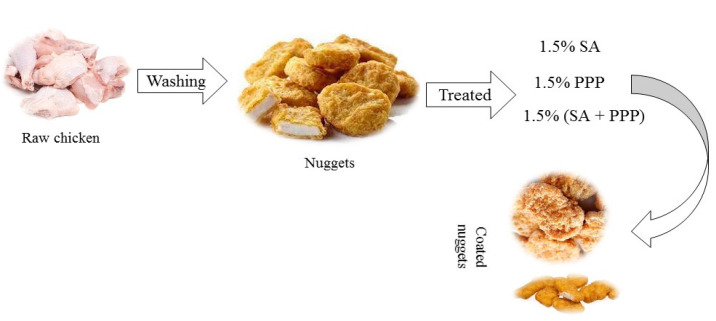
Preparation of chicken pomegranate nuggets coated with different materials.

**Table 1 molecules-27-04500-t001:** Total aerobic count and Coliforms of coated and uncoated chicken nuggets during refrigerated storage.

Treatments	Total Aerobic Count (log CFU/g)					Coliforms (log CFU/g)				
	0	7	14	21	Mean ± SD	0	7	14	21	Mean ± SD
Uncoated	4.23 ± 0.02	4.62 ± 0.04	4.84 ± 0.06	5.09 ± 0.05	4.70 ± 0.08 a	2.42 ± 0.06	3.14 ± 0.06	3.58 ± 0.05	3.91 ± 0.06	3.26 ± 0.05 a
SA (1.5%)	3.97 ± 0.05	4.23 ± 0.07	4.51 ± 0.08	4.64 ± 0.08	4.34 ± 0.04 b	1.92 ± 0.05	2.16 ± 0.04	2.39 ± 0.04	2.58 ± 0.04	2.26 ± 0.07 b
PPP (1.5%)	3.63 ± 0.06	3.79 ± 0.01	3.86 ± 0.04	4.07 ± 0.07	3.84 ± 0.05 c	1.63 ± 0.05	1.88 ± 0.05	1.97 ± 0.06	2.16 ± 0.05	1.91 ± 0.09 c
SA (1.5%) + PPP (1.5%)	3.05 ± 0.01	3.29 ± 0.21	3.48 ± 0.05	3.69 ± 0.06	3.38 ± 0.06 d	1.46 ± 0.04	1.59 ± 0.04	1.69 ± 0.07	1.74 ± 0.02	1.62 ± 0.02 d

SA: Sodium alginate, PPP: pomegranate peel powder. The values are mean ± SD of three independent determinations. Means carrying different letters in columns differ significantly (*p* < 0.05).

**Table 2 molecules-27-04500-t002:** Thiobarbituric acid reactive substances (TBARS) and Peroxide value (POV) of coated and uncoated chicken nuggets during refrigerated storage.

Treatments	TBARS (MDA/kg)					POV (Meq Peroxide/kg)				
	0	7	14	21	Mean ± SD	0	7	14	21	Mean ± SD
Uncoated	0.91 ± 0.02	1.24 ± 0.04	1.49 ± 0.06	1.62 ± 0.03	1.32 ± 0.03 a	0.61 ± 0.06	0.73 ± 0.04	0.84 ± 0.08	0.92 ± 0.03	0.78 ± 0.03 a
SA (1.5%)	0.81 ± 0.05	0.95 ± 0.05	1.11 ± 0.08	1.23 ± 0.04	1.03 ± 0.04 b	0.54 ± 0.04	0.62 ± 0.06	0.71 ± 0.04	0.79 ± 0.03	0.67 ± 0.05 b
PPP (1.5%)	0.68 ± 0.03	0.74 ± 0.02	0.83 ± 0.05	0.93 ± 0.05	0.80 ± 0.05 c	0.48 ± 0.02	0.56 ± 0.03	0.64 ± 0.05	0.72 ± 0.05	0.60 ± 0.02 c
SA (1.5%) + PPP (1.5%)	0.51 ± 0.04	0.69 ± 0.01	0.77 ± 0.03	0.86 ± 0.02	0.71 ± 0.05 d	0.42 ± 0.03	0.49 ± 0.04	0.55 ± 0.01	0.63 ± 0.01	0.52 ± 0.04 d

SA: Sodium alginate, PPP: pomegranate peel powder, MDA: malondialdehyde. The values are mean ± SD of three independent determinations. Means carrying different letters in columns differ significantly (*p* < 0.05).

**Table 3 molecules-27-04500-t003:** Total phenolic content (TPC) and 2, 2-diphenyl-1-picrylhydrazyl (DPPH) inhibition of coated and uncoated chicken nuggets during refrigerated storage.

Treatments	Total Phenolic Contents (mg GAE/g)					DPPH (%)				
	0	7	14	21	Mean ± SD	0	7	14	21	Mean ± SD
Uncoated	112.36 ± 2.03	104.87 ± 2.31	97.33 ± 2.01	91.38 ± 2.15	101.49 ± 2.55 d	55.34 ± 1.00	52.32 ± 1.51	47.88 ± 1.28	43.48 ± 1.06	49.76 ± 1.40 d
SA (1.5%)	119.31 ± 2.05	109.27 ± 2.35	102.54 ± 1.50	96.37 ± 2.25	106.87 ± 2.75 c	58.97 ± 1.05	55.76 ± 1.67	52.53 ± 1.51	49.63 ± 1.37	54.22 ± 1.50 c
PPP (1.5%)	126.71 ± 3.05	115.67 ± 3.01	109.46 ± 1.35	101.37 ± 2.35	113.30 ± 3.13 b	61.92 ± 2.00	58.63 ± 1.75	54.26 ± 1.60	52.49 ± 1.51	56.83 ± 1.60 b
SA (1.5%) + PPP (1.5%)-	135.66 ± 3.07	127.64 ± 3.02	121.74 ± 3.15	117.68 ± 2.45	125.68 ± 3.17 a	64.65 ± 2.15	62.85 ± 1.04	59.57 ± 1.86	56.68 ± 1.72	60.94 ± 170 a

SA: Sodium alginate, PPP: pomegranate peel powder, GAE: Gallic acid equivalents. The values are mean ± SD of three independent determinations. Means carrying different letters in columns differ significantly (*p* < 0.05).

**Table 4 molecules-27-04500-t004:** L* and a* values of coated and uncoated chicken nuggets during refrigerated storage.

Treatments	L*					a*				
	0	7	14	21	Mean ± SD	0	7	14	21	Mean ± SD
Uncoated	58.12 ± 1.06	58.26 ± 1.50	59.63 ± 1.14	60.12 ± 1.96.	59.03 ± 1.84 d	11.36 ± 0.45	11.63 ± 0.47	11.99 ± 0.48	12.03 ± 0.50	11.75 ± 0.47 d
SA (1.5%)	58.36 ± 1.08	58.26 ± 1.60	59.36 ± 1.12	60.96 ± 2.03	59.24 ± 1.85 c	12.63 ± 0.53	12.56 ± 0.51	12.69 ± 0.52	12.89 ± 0.53	12.69 ± 0.52 c
PPP (1.5%)	59.12 ± 1.09	60.36 ± 2.00	61.48 ± 2.01	61.22 ± 2.02	60.55 ± 2.16 b	13.85 ± 0.58	13.89 ± 0.58	13.93 ± 0.58	14.25 ± 0.60	13.98 ± 0.58 b
SA (1.5%) + PPP (1.5%)	61.36 ± 2.01	61.96 ± 2.03	62.02 ± 2.05	62.36 ± 2.15	61.93 ± 2.02 a	14.52 ± 0.61	14.64 ± 0.62	14.69 ± 0.62	14.87 ± 0.63	14.68 ± 0.62 a

SA: Sodium alginate, PPP: pomegranate peel powder. The values are mean ± SD of three independent determinations. Means carrying different letters in columns differ significantly (*p* < 0.05).

**Table 5 molecules-27-04500-t005:** b* and pH values of coated and uncoated chicken nuggets during refrigerated storage.

Treatments	b*					pH				
	0	7	14	21	Mean ± SD	0	7	14	21	Mean ± SD
Uncoated	9.15 ± 0.34	9.45 ± 0.36	9.65 ± 0.37	9.98 ± 0.35	9.56 ± 0.17 d	6.1 ± 0.30	6.17 ± 0.15	6.24 ± 0.20	6.45 ± 0.34	6.24 ± 0.23 a
SA (1.5%)	9.78 ± 0.37	10.36 ± 0.40	10.78 ± 0.45	10.96 ± 0.24	10.47 ± 0.22 c	5.92 ± 0.10	6.12 ± 0.13	6.19 ± 0.17	6.35 ± 0.24	6.15 ± 0.12 d
PPP (1.5%)	10.45 ± 0.41	10.65 ± 0.43	10.78 ± 0.42	10.89 ± 0.13	10.69 ± 0.02 b	5.96 ± 0.13	6.1 ± 0.34	6.22 ± 0.20	6.4 ± 0.05	6.17 ± 0.03 c
SA (1.5%) +PPP(1.5%)	10.58 ± 0.42	10.65 ± 0.44	10.99 ± 0.48	11.25 ± 0.34	10.87 ± 0.24 a	5.89 ± 0.01	6.15 ± 0.17	6.29 ± 0.25	6.42 ± 0.29	6.19 ± 0.08 b

SA: Sodium alginate, PPP: pomegranate peel powder. The values are mean ± SD of three independent determinations. Means carrying different letters in columns differ significantly (*p* < 0.05).

**Table 6 molecules-27-04500-t006:** Appearance and texture of coated and uncoated chicken nuggets during refrigerated storage.

Treatments	Appearance					Texture				
	0	7	14	21	Mean ± SD	0	7	14	21	Mean ± SD
Uncoated	7.89 ± 0.33	6.79 ± 0.32	6.11 ± 0.03	5.98 ± 0.38	6.69 ± 0.28 a	8.12 ± 0.30	7.05 ± 0.02	6.78 ± 0.33	6.1 ± 0.02	7.01 ± 0.02 a
SA (1.5%)	7.15 ± 0.11	6.36 ± 0.12	6.38 ± 0.15	6.01 ± 0.02	6.48 ± 0.17 b	7.49 ± 0.43	6.79 ± 0.32	6.19 ± 0.08	5.79 ± 0.29	6.57 ± 0.21 c
PPP (1.5%)	7.02 ± 0.01	6.65 ± 0.22	6.01 ± 0.01	5.29 ± 0.10	6.24 ± 0.09 c	7.06 ± 0.05	6.79 ± 0.35	6.49 ± 0.32	6.02 ± 0.03	6.59 ± 0.25 b
SA (1.5%) + PPP (1.5%)	7.36 ± 0.14	6.29 ± 0.28	5.89 ± 0.23	5.55 ± 0.17	6.27 ± 0.11 d	7.89 ± 0.35	6.49 ± 0.20	6.16 ± 0.05	5.49 ± 0.15	6.51 ± 0.19 d

SA: Sodium alginate, PPP: pomegranate peel powder. The values are mean ± SD of three independent determinations. Means carrying different letters in columns differ significantly (*p* < 0.05).

**Table 7 molecules-27-04500-t007:** Taste and odor of coated and uncoated chicken nuggets during refrigerated storage.

Treatments	Taste					Odor				
	0	7	14	21	Mean ± SD	0	7	14	21	Mean ± SD
Uncoated	7.75 ± 0.27	7.12 ± 0.22	6.25 ± 0.20	5.68 ± 0.10	6.70 ± 0.22 a	7.48 ± 0.26	6.28 ± 0.20	6.02 ± 0.20	5.72 ± 0.04	6.38 ± 0.24 d
SA (1.5%)	7.46 ± 0.26	6.49 ± 0.20	6.25 ± 0.21	5.79 ± 0.13	6.50 ± 0.13 c	7.16 ± 0.21	6.49 ± 0.30	6.1 ± 0.21	5.9 ± 0.06	6.41 ± 0.27 c
PPP (1.5%)	7.16 ± 0.24	6.89 ± 0.23	6.49 ± 0.23	5.89 ± 0.15	6.61 ± 0.18 b	7.54 ± 0.30	7.06 ± 0.23	6.59 ± 0.25	6.15 ± 0.22	6.84 ± 0.29 a
SA (1.5%) + PPP (1.5%)	7.25 ± 0.25	6.79 ± 0.21	6.12 ± 0.17	5.76 ± 0.12	6.48 ± 0.12 d	7.2 ± 0.20	6.82 ± 0.51	6.19 ± 0.23	5.88 ± 0.06	6.52 ± 0.21 b

SA: Sodium alginate, PPP: pomegranate peel powder. The values are mean ± SD of three independent determinations. Means carrying different letters in columns differ significantly (*p* < 0.05).

**Table 8 molecules-27-04500-t008:** Overall acceptability of coated and uncoated chicken nuggets during refrigerated storage.

Treatments	Overall Acceptability				
	0	7	14	21	Mean ± SD
Uncoated	7.69 ± 0.27	7.02 ± 0.25	6.58 ± 0.24	6.2 ± 0.27	6.87 ± 0.24 a
SA (1.5%)	7.36 ± 0.26	6.49 ± 0.24	6.03 ± 0.21	5.82 ± 0.20	6.43 ± 0.23 b
PPP (1.5%)	7.05 ± 0.25	6.68 ± 0.24	6.05 ± 0.21	5.62 ± 0.20	6.35 ± 0.23 c
SA (1.5%) + PPP (1.5%)	6.99 ± 0.24	6.16 ± 0.22	5.76 ± 0.20	5.69 ± 0.20	6.15 ± 0.22 d

SA: Sodium alginate, PPP: pomegranate peel powder. The values are mean ± SD of three independent determinations. Means carrying different letters in columns differ significantly (*p* < 0.05).

## Data Availability

Not available.

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
