# Peer review of "Effect of Antimicrobial and Antioxidant Rich Pomegranate Peel Based Edible Coatings on Quality and Functional Properties of Chicken Nuggets"

_molecules, 2022, doi:10.3390/molecules27144500_

Round 1
Reviewer 1 Report
Although what is proposed is an interesting one for the use of sub-products from tropical fruits, the document presents a considerable number of points that should be improved.
-Extensive checking of spelling and grammar throughout the text is necessary
-The methodology is not well developed, and the reason for these concentrations to carry out the coat is not established. The methodologies for determining phenolic and microbiological compounds are not clear.
-The brands, models, etc. of the equipment used are not established
Abbreviations for minutes, hours, etc. are not correctly used.
In sensory evaluation, 10 judges are used. Are they trained? If they are, please specify. If they are not, the number of evaluators used will not be able to validate the results presented.
It only mentions that the values of L a and b vary, but it does not explain why this happens, something similar happens with the other results which are not explained in detail.
Author Response
DETAILED RESPONSE TO REVIEWERS’ COMMENTS
The Reviewers’ comments were constructive since these improved the overall quality of our study. There has been an effort to cover adequately all comments.
Response to Reviewer’s 1 Comments
- Although what is proposed is an interesting one for the use of sub-products from tropical fruits, the document presents a considerable number of points that should be improved.
Response: We appreciate the reviewer for the relevant information. The points have been improved significantly in the revised manuscript.
- -Extensive checking of spelling and grammar throughout the text is necessary
Response: There has been an effort to improve the spelling and grammar of the full text article.
- -The methodology is not well developed, and the reason for these concentrations to carry out the coat is not established. The methodologies for determining phenolic and microbiological compounds are not clear.
Response: The methodology has been improved according to the suggestion of the reviewer. However, the methodologies for determining phenolic and microbiological compounds have been rewritten.
- -The brands, models, etc. of the equipment used are not established
Response: Corrected
- Abbreviations for minutes, hours, etc. are not correctly used.
Response: Action taken according to the suggestion of the reviewer.
- In sensory evaluation, 10 judges are used. Are they trained? If they are, please specify. If they are not, the number of evaluators used will not be able to validate the results presented.
Response: The panelists are trained enough to fulfil the criteria according to ASTM standards. Although, the number is less but these are trained panelists.
- It only mentions that the values of L a and b vary, but it does not explain why this happens, something similar happens with the other results which are not explained in detail.
Response: Improvements have been made in the revised manuscript.
Reviewer 2 Report
I reviewed the manuscript entitled, Development of Antimicrobial and Antioxidants Potentials On the Basis of Functional Chicken Nuggets Treated with Pomegranate Peel Based Edible Coating. The manuscript has less novelty. For example, authors performed TPC and TFC and claimed it as a potential antioxidant. The experiments have many flaws.
There are no details on how authors fried the nuggets (time).
What is the mixer of water and other flour ingredients? Frying time and conditions.
Lines 120 and 121: lines are repeated?
In combined SA and PPP, how much for each ?
What about oil-uptake studies? Authors should perform this
Sensory evaluation: are panels members experts? The number is very less
None of the Tables appear with statistical info. Without statistical info, authors cannot claim their hypothesis.
Interpretation and quality of the findings are very poor
References should be according to journal format.
Conclusions should be concise and highlight the experimental findings.
Author Response
DETAILED RESPONSE TO REVIEWERS’ COMMENTS
The Reviewers’ comments were constructive since these improved the overall quality of our study. There has been an effort to cover adequately all comments.
Response to Reviewer’s 2 Comments
- I reviewed the manuscript entitled, Development of Antimicrobial and Antioxidants Potentials On the Basis of Functional Chicken Nuggets Treated with Pomegranate Peel Based Edible Coating. The manuscript has less novelty. For example, authors performed TPC and TFC and claimed it as a potential antioxidant. The experiments have many flaws.
Response: We performed TPC and DPPH for the claim of potential antioxidant activity. Furthermore, all flaws have been improved significantly in the revised manuscript.
- There are no details on how authors fried the nuggets (time).
Response: Chicken nuggets fried detail have been added in the revised manuscript.
- What is the mixer of water and other flour ingredients? Frying time and conditions.
Response: The mixture detail and frying condition have been added according to the suggestion of the reviewer.
- Lines 120 and 121: lines are repeated?
Response: Changes have been made in the revised manuscript.
- In combined SA and PPP, how much for each ?
Response: The combination of SA and PPP was 1.5% and 1.5% respectively
- What about oil-uptake studies? Authors should perform this
Response: The oil-uptake studies have not been performed in this manuscripts.
- Sensory evaluation: are panels members experts? The number is very less
Response: The panelists are trained enough to fulfil the criteria according to ASTM standards. Although, the number is less but these are trained panelists.
- None of the Tables appear with statistical info. Without statistical info, authors cannot claim their hypothesis.
Response: All the tables in this manuscript have been appeared with statistical manner
- Interpretation and quality of the findings are very poor
Response: Interpretation and quality have been improved according to the suggestion of the reviewer.
- References should be according to journal format.
Response: Action taken according to the suggestion of the reviewer.
- Conclusions should be concise and highlight the experimental findings.
Response: Changes have been made according to the suggestion of the reviewer.
Round 2
Reviewer 1 Report
Line 109-112. in the development of the formulation of the nuggets please separate the g from the numbers
Line 106. electric mincer (brand, model, city, etc)
Line 147. About seconds the correct abbreviation should be s not sec
Line 156-158. Please clarify below to the equation.. m=?, mo=? M the value 55.58 what means?
Line 178 Calibration curve. . What is "0 mg/mL"? The curve can be extrapolated through zero but not 0 µg/mL concentration. Please correct. Or explain.
Line 183. 1h
Line 200. Although you have already told me that your panel has a certain degree of training, please clarify it in the text, so that anyone who reads it will understand why only 10 panelists were used in a hedonic test.
Line 129-131.Are you sure about this?????
Author Response
DETAILED RESPONSE TO REVIEWERS’ COMMENTS
The Reviewers’ comments were constructive again since these improved the overall quality of our study. There has been an effort to respond and cover adequately all comments.
Response to Reviewer’s 1 Comments
Reviewer 1
Line 109-112. in the development of the formulation of the nuggets please separate the g from the numbers
Response: Action taken according to the suggestion of the reviewer.
Line 106. electric mincer (brand, model, city, etc)
Response: Added.
Line 147. About seconds the correct abbreviation should be s not sec
Response: The correct abbreviation of seconds has been added according to the suggestion of the reviewer.
Line 156-158. Please clarify below to the equation.. m=?, mo=? M the value 55.58 what means?
Response: Action taken according to the suggestion of the reviewer.
Line 178 Calibration curve. . What is "0 mg/mL"? The curve can be extrapolated through zero but not 0 µg/mL concentration. Please correct. Or explain.
Response: The correction has been made according to the suggestion of the reviewer.
Line 183. 1h
Response: Corrected
Line 200. Although you have already told me that your panel has a certain degree of training, please clarify it in the text, so that anyone who reads it will understand why only 10 panelists were used in a hedonic test.
Response: The changes have been made according to the suggestion of the reviewer.
Line 129-131.Are you sure about this?????
Response: Yes
Reviewer 2 Report
Although the authors improved the manuscript against the queries raised by the reviewers, but the manuscript still need essential changes.
Major:
1. Lack of novelty : if the antioxidant-extract of pomegranate peel powder will be selected for the coating, then the study could give significant positive results, now only one log reductions in coated samples.
The direct powder addition had not shown much significance difference.
Now a days nanocoating or nano-packaging in the trend, the authors did not see towards the novelty aspects.
2. After frying (180 °C), there is question of stability of antioxidants.
3. Nutritional composition of the new product is missing
4. The film forming ability and others characteristics of the coating materials should be considered.
Therefore, I suggest to the rejection.
Here are some suggestions to improve the manuscript:
1. Line no. 111…………..Egg should be egg
2. Line no. 117…………..where you previously mentioned……….
3. Line no. 120…..mention all the coating compositions not only about SA. In figure, all the coating compositions mentioned…so mention in text as well
4. Line no. 121…..you are not supposed to use the word melt…SA is always in powder form, modify your sentence
5. Line no. 122-124……..A plasticizer agent (glycerine) was added and then the hydrolysed solution was cooled at 70 °C and the 1.5% solution of PPP was also added. The coated solution was used for the chicken nuggets; then, these were dipped in the 2% solution of salt (CaCl2) for 1 minute.
Revise the sentence and as you mentioned you prepared alone SA, and how you developed PPP (with and without SA), explain alone SA and alone PPP.
6. Also mention the concentration of plasticizer used
7. Revise the line 148-149…
Briefly, 50 μL of butylated hydroxyanisole (BHA) 7.2% and 2mL of 20 mM solution of thiobarbituric acid (TBA) in 15% trichloroacetic acid (TCA) solution were combined in a test tube.
8. Line no. 153-154…………what do you measure from the following equation?
“TBARS values were estimated by measuring the absorbance at 532 nm using a spectrophotometer (Milton Roy Co., Rochester, NY) and the following equation:”
Please revise the sentence with significance (As mentioned in line 163-164).
9. Line no. 157-158……..insert in equation format…according to manuscript standards, also correct: mg of malonic dialdehyde per kg meat……..as “Malonic dialdehyde (mg/ kg meat)”
10. Line no. 161…………revise this as…..according to the method suggested by International Dairy Federation (IDF),..
11. Line 166………all equations should be in the equation format with eq. numbers…..please follow the manuscript standards
12. Line 172: “The samples prepared from 125 μL nuggets solution,…which samples you prepared”……kindly revise the line….It seems nuggets are in a solution form, how you extracted the polyphenols from nuggets?
13. Line 186………all equations should be in the equation format with eq. numbers…..please follow the manuscript standards
14. Revise the line 218-110: “The overall mean values of total aerobic count of uncoated, SA (1.5 %), PPP (1.5 %), and SA (1.5 %) + PPP (1.5 %) at refrigerated temperature for 0, 7, 14, and 21 days were 3.38 ± 0.06, 3.84 ± 0.05, 4.34 ± 0.04, and 4.70 ± 0.08 log CFU/g, respectively.”
Here…..three coatings (SA (1.5 %), PPP (1.5 %), and SA (1.5 %) + PPP (1.5 %)) and four variable of storage days (0, 7, 14, and 21 days)……how you can mention only four values (3.38 ± 0.06, 3.84 ± 0.05, 4.34 ± 0.04, and 4.70 ± 0.08 log CFU/g, respectively), these values may be of one coating with four different days….or how?
15. Line no. 222: The microbial population for total aerobic count ranged between 3.05 ± 0.01 to 4.23 ± 0.02 log CFU/g at the initial stage of the experiment…..was it for 0th day? Please clarify
16. Line 226: An extreme value of total aerobic count (5.09 ± 0.05 log CFU/g) was recorded on the 21 day of storage,…..in which sample?
17. Line 227: whereas the least value of total aerobic count (3.05 ± 0.01 log CFU/g) was obtained with the coating of SA (1.5%) and PPP (1.5%) on day 0th…….was it much effective on the 0th day?
18. Please write 0th , 21st day………likewise
19. Line 236-237: please revise the line as mentioned above in my query point number 14.
20. Line 273-274: The mean values of uncoated, SA (1.5 %), PPP (1.5 %), and SA (1.5 %) + PPP (1.5 %) formulations at refrigerated temperature for 0, 7, 14, 21 days were 0.71 ± 0.05, 0.80 ± 0.05, 1.03 ± 0.04, and 1.32 ± 0.03 MDA/kg, respectively. There should be 16 values if you mentioning here all….four coated treatments…with four days…… Please revise these kind of sentences throughout your manuscript…..
21. Please follow the same in line : 291-293
22. Please revise your manuscript by reading carefully, it lacking the writing quality standards and confusing the readers.
Author Response
DETAILED RESPONSE TO REVIEWERS’ COMMENTS
The Reviewers’ comments were constructive again since these improved the overall quality of our study. There has been an effort to respond and cover adequately all comments.
Response to Reviewer’s 2 Comments
Reviewer 2
Comments and Suggestions for Authors
Although the authors improved the manuscript against the queries raised by the reviewers, but the manuscript still need essential changes.
Major:
- Lack of novelty: if the antioxidant-extract of pomegranate peel powder will be selected for the coating, then the study could give significant positive results, now only one log reductions in coated samples.
The direct powder addition had not shown much significance difference.
Now a days nanocoating or nano-packaging in the trend, the authors did not see towards the novelty aspects.
Response: Dear reviewer, thank you for your concern. Actually, our main objective was the use of agricultural waste and the value addition. Secondly, the antioxidant profile and stability of chicken nuggets was also significantly improved by using the pomegranate peel powder. So, overall, it’s the novelty by using the natural coating material with the agricultural waste. The reviewer is also right regarding the nanocoating. In our future articles, we are working on the nanocoatings as well.
- After frying (180 °C), there is question of stability of antioxidants.
Response: The minor changes were observed of stability of antioxidants during study
- Nutritional composition of the new product is missing
Response: As we discussed earlier, our major focus was on the stability, antioxidant profile and microbial assay and definitely, we are working on the nutritional composition in our other article.
- The film forming ability and others characteristics of the coating materials should be considered.
Response: The nutritional, film forming ability and some analytical parameters are in progress for another article.
Therefore, I suggest to the rejection.
Here are some suggestions to improve the manuscript:
- Line no. 111…………..Egg should be egg
Response: The word “Egg” has been replaced with “egg”.
- Line no. 117…………..where you previously mentioned……….
Response: The correction has been made.
- Line no. 120…..mention all the coating compositions not only about SA. In figure, all the coating compositions mentioned…so mention in text as well
Response: The changes have been made according to the suggestion of the reviewer.
- Line no. 121…..you are not supposed to use the word melt…SA is always in powder form, modify your sentence
Response: Action taken according to the suggestion of the reviewer.
- Line no. 122-124……..A plasticizer agent (glycerine) was added and then the hydrolysed solution was cooled at 70 °C and the 1.5% solution of PPP was also added. The coated solution was used for the chicken nuggets; then, these were dipped in the 2% solution of salt (CaCl2) for 1 minute.
Revise the sentence and as you mentioned you prepared alone SA, and how you developed PPP (with and without SA), explain alone SA and alone PPP.
Response: The sentence has been revised according to the suggestion of the reviewer.
- Also mention the concentration of plasticizer used
Response: Action taken according to the suggestion of the reviewer.
- Revise the line 148-149…
Briefly, 50 μL of butylated hydroxyanisole (BHA) 7.2% and 2mL of 20 mM solution of thiobarbituric acid (TBA) in 15% trichloroacetic acid (TCA) solution were combined in a test tube.
Response: The lines have been revised according to the suggestion of the reviewer.
- Line no. 153-154…………what do you measure from the following equation?
“TBARS values were estimated by measuring the absorbance at 532 nm using a spectrophotometer (Milton Roy Co., Rochester, NY) and the following equation:”
Please revise the sentence with significance (As mentioned in line 163-164).
Response: The sentence has been revised according to the suggestion of the reviewer.
- Line no. 157-158……..insert in equation format…according to manuscript standards, also correct: mg of malonic dialdehyde per kg meat……..as “Malonic dialdehyde (mg/ kg meat)”
Response: Action taken according to the suggestion of the reviewer.
- Line no. 161…………revise this as…..according to the method suggested by International Dairy Federation (IDF),..
Response: The lines have been revised according to the suggestion of the reviewer.
- Line 166………all equations should be in the equation format with eq. numbers…..please follow the manuscript standards
Response: Action taken according to the suggestion of the reviewer.
- Line 172: “The samples prepared from 125 μL nuggets solution,…which samples you prepared”……kindly revise the line….It seems nuggets are in a solution form, how you extracted the polyphenols from nuggets?
Response: The lines have been revised according to the suggestion of the reviewer.
- Line 186………all equations should be in the equation format with eq. numbers…..please follow the manuscript standards
Response: Action taken according to the suggestion of the reviewer.
- Revise the line 218-110:“The overall mean values of total aerobic count of uncoated, SA (1.5 %), PPP (1.5 %), and SA (1.5 %) + PPP (1.5 %) at refrigerated temperature for 0, 7, 14, and 21 days were 3.38 ± 0.06, 3.84 ± 0.05, 4.34 ± 0.04, and 4.70 ± 0.08 log CFU/g, respectively.”
Here…..three coatings (SA (1.5 %), PPP (1.5 %), and SA (1.5 %) + PPP (1.5 %)) and four variable of storage days (0, 7, 14, and 21 days)……how you can mention only four values (3.38 ± 0.06, 3.84 ± 0.05, 4.34 ± 0.04, and 4.70 ± 0.08 log CFU/g, respectively), these values may be of one coating with four different days….or how?
Response: Corrections have been made according to the suggestion of the reviewer.
- Line no. 222: The microbial population for total aerobic count ranged between 3.05 ± 0.01 to 4.23 ± 0.02 log CFU/g at the initial stage of the experiment…..was it for 0thday? Please clarify
Response: Changes have been made according to the suggestion of the reviewer.
- Line 226: An extreme value of total aerobic count (5.09 ± 0.05 log CFU/g) was recorded on the 21 day of storage,…..in which sample?
Response: Data about extreme value of total aerobic count has been added.
- Line 227: whereas the least value of total aerobic count (3.05 ± 0.01 log CFU/g) was obtained with the coating of SA (1.5%) and PPP (1.5%) on day 0th…….was it much effective on the 0thday?
Response: Yes, it was much effective on the 0th day
- Please write 0th, 21st day………likewise
Response: Action taken according to the suggestion of the reviewer.
- Line 236-237: please revise the line as mentioned above in my query point number 14.
Response: The sentence has been revised according to the suggestion of the reviewer.
- Line 273-274: The mean values of uncoated, SA (1.5 %), PPP (1.5 %), and SA (1.5 %) + PPP (1.5 %) formulations at refrigerated temperature for 0, 7, 14, 21 days were 0.71 ± 0.05, 0.80 ± 0.05, 1.03 ± 0.04, and 1.32 ± 0.03 MDA/kg, respectively. There should be 16 values if you mentioning here all….four coated treatments…with four days…… Please revise these kind of sentences throughout your manuscript…..
Response: The sentences have been revised according to the suggestion of the reviewer.
- Please follow the same in line : 291-293
Response: Changes been made according to the suggestion of the reviewer.
- Please revise your manuscript by reading carefully, it lacking the writing quality standards and confusing the readers.
Response: Action taken according to the suggestion of the reviewer.